# Waning of SARS-CoV-2 Seropositivity among Healthy Young Adults over Seven Months

**DOI:** 10.3390/vaccines10091532

**Published:** 2022-09-15

**Authors:** C. Suzanne Lea, Kristina Simeonsson, Aaron M. Kipp, Charleen McNeill, Lisa Wilcox, William Irish, Hannah Morris, Omar M. Diaz, John T. Fallon, Rachel L. Roper

**Affiliations:** 1Department of Public Health, Brody School of Medicine, East Carolina University, Greenville, NC 27834, USA; 2Department of Pediatrics, Brody School of Medicine, East Carolina University, Greenville, NC 27834, USA; 3College of Nursing, East Carolina University, Greenville, NC 27834, USA; 4Department of Surgery, Brody School of Medicine, East Carolina University, Greenville, NC 27834, USA; 5Department of Pathology, Brody School of Medicine, East Carolina University, Greenville, NC 27834, USA; 6Department of Microbiology and Immunology, Brody School of Medicine, East Carolina University, Greenville, NC 27834, USA

**Keywords:** severe acute respiratory syndrome coronavirus 2 (SARS-CoV-2), seroprevalence, young adults, longitudinal, nucleocapsid protein, spike protein

## Abstract

Background: We conducted a longitudinal study to estimate immunity produced in response to severe acute respiratory syndrome coronavirus 2 (SARS-CoV-2) infection among university students over seven months. Methods: All participants were attending a public university and resided in Pitt County, North Carolina. University students enrolled weekly for 10 weeks between 26 August 2020 and 28 October 2020, resulting in 136 young adults completing at least one study visit by 17 November 2020. Enrolled students completed an online survey and nasal swab collection at two-week intervals and monthly blood collection between 26 August 2020 and 31 March 2021. Results: Amongst 695 serum samples tested during follow-up, the prevalence of a positive result for anti-nucleocapsid antibodies (N-IgG) was 9.78%. In 22 students with more than one positive N-IgG serum sample, 68.1% of the group lost persistence of N-IgG below the positive threshold over 140 days. Anti-spike IgG antibodies were significantly higher among 11 vaccinated compared to 10 unvaccinated. Conclusions: In healthy young adults, N-IgG wanes below the detectable threshold within five months. S-IgG titer remained consistently elevated months after infection, and significantly increased after vaccination.

## 1. Introduction

Severe acute respiratory syndrome coronavirus-2 (SARS-CoV-2), which causes coronavirus disease 19 (COVID-19), emerged in China in late 2019 [1]. SARS-CoV-2 belongs to the *Betacoronavirus* genus in the family *Coronaviridae*. The virion particle contains four primary structural virion proteins: nucleocapsid (N), spike (S), envelope (E), and membrane (M).

In the fall semester of 2020, students enrolled in US colleges and universities returned to campuses for in-person learning and residential living. Most US colleges and universities offered COVID-19 testing to symptomatic students and to asymptomatic contacts of confirmed cases [2,3]. Testing asymptomatic persons in a congregate setting has been part of a comprehensive strategy to reduce transmission [4,5], since young adults may spread the infection while asymptomatic [6,7,8].

Measuring specific antibodies to SARS-CoV-2 is another way to monitor virus prevalence and spread. Early in the pandemic, the common target antigens for serological assays were the nucleocapsid (N) protein and the spike (S) protein. The presence of anti-SARS-CoV-2 immunoglobin IgG (IgG) indicates the individual has been infected and mounted an immune response to the virus from current or prior infection. Among PCR positives, seroprevalence ranges from 88–100% in large studies [9,10,11]. A vast majority of SARS-CoV-2 infected individuals seroconvert for a duration of months [10], and natural immunity may persist for longer than 12 months [12]. In persons with asymptomatic or mild cases, IgG seroconversion takes a longer time to mount, and the peak antibody response is lower than in those with more serious systemic disease [13,14,15]. IgG antibody responses wane over time [9,16,17,18], particularly nucleocapsid antibodies [15,19]. Importantly, the strength and longevity of the antibody response inform whether persons are likely to be protected from reinfection [10,20,21].

Longitudinal studies have been conducted to examine the persistence of IgG antibody duration over time, mostly in healthcare workers [18,22,23], long-term care residents [19], and COVID-19 patients [9,10,13,24,25,26]. Limited longitudinal evidence exists on seroconversion and reversion among healthy young adults enrolled at a university. Reports suggest that peak antibody levels are lower in those with asymptomatic to mild infection [9,15,16], and a rapid decline of IgG immunity has been documented in those who have recovered from COVID-19 [9,16,21,25]. Less is understood about immunity persistence in healthy young adults.

We conducted a surveillance research program among a cohort of university students before vaccination was widely available. We hypothesized that IgG status would decline over time, consistent with data from healthcare workers. Herein, we report the seroprevalence of N-IgG, the time to N-IgG development, and the occurrence of seroreversion for N-IgG, including change in antibodies to Spike protein (S-IgG) among a group of university students.

## 2. Materials and Methods

### 2.1. Participant Recruitment and Eligibility

University and Medical Center Institutional Review Board (UMCIRB) approved the research (#20-002665). As part of reopening a large public university in North Carolina (NC) (East Carolina University, ECU, Greenville, NC, USA) during fall semester 2020, we implemented a surveillance research project that included testing a group of students bi-monthly for presence of active SARS-CoV-2 infection and monthly for development of humoral immunity. In spring semester 2021, this same cohort was invited once per month for three consecutive months to test for active infection and immunity. An undergraduate or graduate student was eligible if enrolled full or part-time in fall semester 2020, listed a Pitt County residential address, was 18 years of age or older, and spoke English. Students affiliated with the ECU athletics program were excluded due to testing under a separate protocol.

### 2.2. Sampling and Recruitment

Nine hundred students were randomly selected for enrollment. Due to multiple large clusters of COVID-19 disease among the student body, closure of university dormitories to most students beginning 26 August 2020 eliminated a majority of randomly selected students from eligibility. Additional students were recruited during September and October through announcements posted on university social media pages and distributed to student organizations and departmental rosters (e.g., dance, theater, art). During fall semester 2020, students were enrolled weekly from 20 August 2020 until 28 October 2020 (Wave 1). Any participant with at least one study visit during Wave 1 was invited to continue during spring semester 2021 to complete a study visit on 27 January, 24 February, and 31 March 2021 (Wave 2).

### 2.3. Data Collection

Participants completed an online survey, which included symptoms, behaviors, and beliefs every two weeks concurrent with each collection visit during fall 2020 and once monthly during spring 2021. A $20 incentive was provided when the survey and nasopharyngeal (NP) swab collection were completed. The survey was updated to collect information on vaccination vendor (Pfizer/BioNTech, Moderna, Janssen) and vaccination dates in Wave 2. No vaccinations were provided as part of study participation. Study data were collected and managed using REDCap (Research Electronic Data Capture), a secure, web-based software platform, hosted at East Carolina University [27].

### 2.4. Study Visit Procedures

Students received a date and time for the clinic visit, which occurred every two weeks during fall 2020 and once monthly during spring 2021. Signed informed consent was obtained at the student’s first study visit for Wave 1 and Wave 2, respectively. There were 13 study visits in Wave 1 and 3 study visits in Wave 2. If a student missed a scheduled visit, he/she was requested to attend the next week’s study visit.

Serology. After collection of the NP swab by a physician or nurse, a trained phlebotomist collected approximately 5 mL of venous blood into a serum separator tube using a butterfly needle for analysis of N-IgG. A second tube of blood was collected on a subgroup of 25 individuals re-consented during Wave 2 to identify antibodies to spike protein (S-IgG). The 25 students were: 22 participants who were seropositive for N-IgG during fall 2020 or spring 2021; 2 participants who tested PCR positive during a Wave 1 study visit but were seronegative for N-IgG on 17 November 2020; and 1 who self-reported PCR positive at enrollment but did not test N-IgG positive during Wave 1. No samples were collected between 18 November 2020 and 23 January 2021.

### 2.5. Laboratory Methods

A description of NP swab collection and analysis is available [28].

Antibodies. Laboratory analysis of immunoglobulin G (IgG) antibodies to the SARS-CoV-2 virus was conducted using a chemiluminescent microparticle immunoassay (CMIA) for the qualitative detection of IgG antibodies to the SARS-CoV-2 nucleoprotein (N-IgG). The recommended index value threshold of 1.4 signal-to-cutoff (S/C) ratio or above (≥1.4) was used to define IgG seropositivity [29,30]. We collected blood every 4 weeks to allow for IgG seroconversion, which ranges from 14 days to 4 weeks after symptom onset [21,31]. Antibody titers were not performed on N-IgG blood samples.

Both IgG antibodies to the SARS-CoV-2 spike protein (S-IgG) and nucleocapsid protein (N-IgG) were examined at three time points: 27 January, 24 February, and 31 March 2021. The spike protein, the structural protein often used as a target for characterizing the immune response to SARS-CoV-2, contains the receptor binding domain (RBD) that the virus uses to dock to its cellular receptor, angiotensin-converting enzyme-2 [13].

The S-IgG responses were measured using an enzyme-linked immunosorbent assay (ELISA) as per previous work [32,33,34]. Ninety-six-well ELISA plates (Immulon H2B; Thermo Scientific, Waltham, MA, USA) were coated with 0.09 µg/well of recombinant SARS-CoV-2 spike, (R&D System, Cat #10549-CV, Minneapolis, MN, USA) in coating buffer (pH 9.8) at 4 °C overnight. Plates were blocked with 2% fetal bovine serum in PBS at room temperature for 30 min. Plates were washed twice with ELISA wash buffer (1× PBS, 0.02% Tween 20, 0.1% NaN_3_), and human sera were added with a serial dilution of 1:3. Plates were incubated at room temperature for 2 h and washed 3 times with ELISA wash buffer. Alkaline phosphatase (AP)-conjugated goat anti-human IgG (H + L) (1:2200, Promega, Madison, WI, USA) was added and incubated at room temperature for 2 h. Plates were washed 3 times and developed (alkaline phosphate substrate kit; Bio-Rad, Hercules, CA, USA), and the absorbance was read at 405 nm in AccuSkan FC (Skanlt 6.1, Fisher Scientific, Waltham, MA, USA). A positive result was defined as four-fold increase in optical density absorbance compared to the negative control value (0.06), or 0.24 optical density absorbance. The negative value was set based on result from two participants who tested PCR positive but did not have a positive nucleocapsid IgG at any time during follow-up. The spike protein assay was performed in the lab of Dr. Rachel Roper, PhD (Brody School of Medicine).

### 2.6. Outcome Measures

The primary endpoints were N-IgG persistence and N-IgG seroreversion. Persistence was defined as the number of days between the date of initial positive N-IgG result and the date of last positive N-IgG result when both N-IgG values were positive. Seroreversion (loss of N-IgG) was defined as the number of days between initial positive N-IgG result and the date of first negative N-IgG [35]. N-IgG positivity is defined as S/C ratio ≥ 1.4. Date of PCR positive test to date of initial N-IgG positive was included in results since the initial date of N-IgG positive was used to calculate persistence. Mean change in S-IgG before and after vaccination was compared.

### 2.7. Statistical Analysis

Participant characteristics were summarized by N-IgG status (positive versus not positive) and compared using chi-square test. Missing demographic and baseline data were treated as missing and were not imputed. Duration of N-IgG seropositivity was summarized by N-IgG persistence status by presenting the mean, standard deviation (SD), median, with minimum and maximum. Time to loss of N-IgG persistence was analyzed using the Kaplan–Meier method. Participants with persistent N-IgG (no loss below S/C ratio 1.4) were right censored (coded zero). The N-IgG and S-IgG profiles for participants were graphically displayed using spaghetti plots. Plots were used as guides to assess change in the N-IgG assay over time. Paired t-test compared change in S-IgG status before and after vaccination. Analyses were performed using Excel (Windows 10), GraphPad Prism v9, and Stata 14. All tests were 2-tailed with *P*-value less than 0.05 as statistically significant.

## 3. Results

One hundred and thirty-six consenting students completed at least one clinic visit in Wave 1, of which, 81 were randomly selected and 55 self-selected to participate. During Wave 1, 86 completed all scheduled study visits (63.2%). Ninety-seven participants from Wave 1 reconsented for Wave 2 (71.3%). Twelve students completed all scheduled study visits between 26 August 2020 and 31 March 2021. There were 13 study visits in Wave 1 and 3 study visits in Wave 2. No new positive PCR tests were detected during Wave 2 study visits.

Figure 1 displays enrollment in fall 2020 and spring 2021. Sixteen participants were N-IgG positive during September–October 2020 and 12 were positive during January (*n* = 7), February (*n* = 2), and March (*n* = 3), 2021.

Ninety-four women (69%) and forty-two men (31%) participated in at least one clinic visit during fall 2020 (Appendix A). Seventy-one percent (71.4%) of N-IgG positive were ages 18 to 21 and thirty-two percent were freshman. A majority of N-IgG positive had obtained testing for COVID-19 prior to enrollment compared to N-IgG negative. No differences between groups were found for having a positive COVID-19 test or completing one or two doses of vaccine.

### 3.1. Persistence and Loss of Anti-Nucleocapsid Antibodies

Approximately 9.8% of serum samples tested positive for N-IgG (68/695 [627 + 68 = 695]) between 26 August 2020 and 31 March 2021. The range of N-IgG S/C ratios was 0.01 to 7.01. At the time of blood collection, 28 participants tested N-IgG positive (S/C ≥ 1.4) during the study interval (Figure 2). During Wave 1, the proportion of newly identified positive N-IgG was 11.76% (16/136) and during Wave 2, the proportion for newly identified N-IgG was 12.37% (12/97) (Appendix A). Over 16 study visits, the percentage of positive N-IgG among those scheduled to provide a blood sample ranged from 0.45% to 14.3%. The mean N-IgG levels among any positives ranged from 1.88 to 4.06 (Appendix A).

Almost 44% (10/23) of students had N-IgG measures below the positivity value (S/C ≥ 1.4) at the end of Wave 1 (31 March 2021). Seven new N-IgG positive tests were reported on 27 January 2021, and of those, four lost positive N-IgG status by the end of Wave 2 (31 March 2021) (57.1%). Three tested N-IgG positive for the first time on 31 March 2021, the last scheduled blood collection day.

Twenty-three of twenty-eight students had more than one positive N-IgG result. Persistent N-IgG positivity and loss were estimated in these 23 students (Table 1). Sixty-five percent (15/23) of participants with more than one positive N-IgG test lost seropositivity. The mean number of days between a positive PCR test and the first measure of positive N-IgG antibodies was 21.21 (14.76, SD) days. The mean and median duration of N-IgG persistence was 54.3 and 48 days, respectively (defined as the number of days between the first positive N-IgG result and the last positive N-IgG result, when both measures remained positive) [35].

Figure 3 displays the number of days from the initial N-IgG positive to the last measure of N-IgG positive (column 3 from Table 1, “IgG persistence”) for 22 students. At 140 days (~4.7 months), 68.1% (15/22) of participants had lost persistent N-IgG status (below 1.4 S/C threshold) by 31 March 2021. The mean (SD) and median number of days were 77.8 (9.9) and 62, respectively, with interquartile estimates between 29 days to 105 days. One participant was lost to follow-up in September 2020 and excluded from Figure 3 (remained in Table 1).

### 3.2. Presence of Anti-Spike Antibodies

Ninety-one percent (51/56) of serum samples had detectable anti-spike IgG (S-IgG) antibodies (Appendix A) at any of the three time points (27 January, 24 February, or 31 March 2021). One hundred percent of students who ever had detectable N-IgG were S-IgG positive during Wave 2. A total of 48% (12/25) of students completed each visit in Wave 2 and 66.6% of those completing three visits in Wave 2 (8/12) had at least one dose of vaccine as of 31 March 2021.

Eleven had obtained at least one dose of vaccine and two students had obtained two doses of vaccine. Figure 4 displays the box and whisker plots (Figure 4A,C) and line graphs (Figure 4B,D) for the Wave 2 S-IgG subgroup (Figure 4A,B) and the vaccinated (Figure 4C,D) among this sub-group, respectively. The mean S-IgG titers were not significantly different between 27 January and 24 February (*p* = 0.82) (Appendix A). On 31 March 2021 among the vaccinated (*n* = 11), the mean S-IgG titer was 1.098 compared to 0.5603 among the unvaccinated (*n* = 10) (*p* = 0.003). One student was first vaccinated on 27 January 2021 and completed the second dose three weeks later. Ten students received their first vaccine dose after 2 March 2021 (see Appendix A). Two students received one dose of the Janssen vaccine and the remaining students received mRNA Pfizer/BioNTech.

Two participants (E and Q), who tested PCR positive in the fall of 2020, never had detectable N-IgG and did not have detectable S-IgG during January or February. Participant Q, who was not vaccinated, remained negative for detectable N-IgG and S-IgG in March 2021. For Participant E, who was vaccinated on 30 March 2021, the S-IgG value between 27 February 2021 and 31 March 2021 changed by 331% for participant E.

## 4. Discussion

Repeated measures of serum antibodies in a young adult university cohort serves to inform the capacity for adaptive immunity through antibody persistence and loss. Among a group of 22 young adults who were followed over seven months, we found that N-IgG positive reverted to below the threshold in 68.1% of participants over 140 days. As expected, levels of S-IgG were markedly elevated in those recently vaccinated. A decline in S-IgG also occurred over the months in the unvaccinated, though titers remained above the detection limit in the absence of vaccination. Our results are suggestively consistent with findings that the anti-S-IgG confers more persistence than anti-N-IgG [10,15,19]. This may be due to the initial response to S-IgG being stronger due to its fundamental immunogenicity or the large quantity of S-IgG produced during infection. This is in contrast to N-IgG, which is sequestered inside the virus particle or inside the infected cell. Alternatively, the laboratory specificity of anti-spike antibodies may be higher than for anti-nucleocapsid antibodies. This specificity cannot be directly compared, because the S and N proteins are fundamentally and biochemically different proteins and different reagents are used to detect them.

Longitudinal studies conducted in a non-hospitalized young adult university population that examined immunoglobulin seroconversion and seroreversion were not identified. Several studies have examined seroconversion and loss over five months or more in clinic patients. In a group of COVID-19 patients, Maine and colleagues tested 427 sequential COVID-19 serum samples between March and August 2020 collected up to 168 days post onset of symptoms [25]. The median days from IgG seronegative to seroconversion was 11.5 days, and 10 patients followed for more than 100 days all experienced IgG decline, similar to our findings [25]. Among 45 Belgian COVID-19 cases with mild disease (defined as asymptomatic or not needing hospitalization), 61.1% were seronegative within 6 months after first PCR positive [14]. Similar findings emerged among non-severe patients in the country of Liechtenstein [36]. The median N-IgG S/C ratio declined significantly (S/C ratio 4.2 to 1.8) between 48 and 140 days among 82 non-severe patients during spring 2020 [36]. In two outbreaks at the same long-term care facility, all ten residents in both outbreaks had a significant anti-N antibody decline, with all ten-second serology measures remaining barely positive after seven months; the decline in anti-spike over four months was not significantly different, similar to our findings [19].

In addition, declines in N-IgG were noted among cohorts of healthcare workers. Anna and colleagues found that anti-N titers declined by 31% over 4–8 weeks in the majority of French healthcare workers [18]. In a cohort of hospital staff in Wales, four months after a positive PCR test, only 22% (2/9) had detectable antibodies against nucleocapsid protein, while 78% (7/9) individuals had detectable antibodies against spike protein [23]. In Japan, 18.2% (6/33) of healthcare workers had detectable anti-N IgG 12 months later [37]. Given that 65% of students lost detectable N-IgG over 140 days, our findings are consistent with other studies in non-hospitalized adults. Forty-six percent of our study population was between the ages of 18–20. Our data suggest that young adults who have asymptomatic or mild disease have a rapid peak and more rapid decline in peak N-IgG antibody response than hospitalized patients.

Strengths of this study include the longitudinal design in a healthy young adult university-based population. Compliance with scheduled study visits was high in both study segments (fall semester 2020 and spring semester 2021). Limitations include the absence of S-IgG and titers during Wave 1. Gaps in monthly blood collection (approximately every 30 days) due to some missed appointments may dilute important changes over time. Thirty days between each sera collection may inflate the number of days between persistence or seroreversion. Since the study population was students currently enrolled at our university, we decided against storing sera for future analysis. We did not collect symptoms or disease severity for self-reported COVID-19 disease that occurred prior to enrollment. We did not collect a second tube on all 97 consenting students in spring 2021 due to budgetary constraints and have not collected additional data on re-infection or other measures since March 2021 for the same reasons. This student population participating in this study may not be representative of the ECU student population as a whole or young adults in general. Seropositivity was assessed in students who were permitted to attend in-person courses on campus, and/or live in the dorms, and otherwise, be motivated to travel to campus-based testing for a bi-monthly visit. This study was implemented six months after the pandemic was declared in March 2020, and vaccination was not readily available for this age group. The alpha, beta, and gamma variants were the dominant strains in NC during Wave 1 and Wave 2. As the pandemic has progressed, a greater understanding of IgG serology assays has emerged along with the significant expansion of laboratory capabilities. Reliance on detection of N-IgG antibodies exclusively for determining potential COVID-19 immunity should be used with caution and multiple independent assays may improve the accuracy of estimating seroprevalence over time [15,36,38].

## 5. Conclusions

Healthy young adults mounted an antibody response to N-IgG that waned to below limits of detection in 68% of participants over 140 days. The S-IgG response rose sharply after vaccination. Acquired infection is an important determinant of protection from reinfection, and the detection of antibodies can help identify persons who have had subclinical or asymptomatic disease. N-IgG antibodies were measured early in the pandemic to assess seroprevalence. Our findings support that healthy young adults did not have long-lasting N-IgG levels after infection. Future studies are needed to characterize the occurrence of reinfection and severity of disease in healthy young adults with emerging COVID-19 strains, including the kinetics of N-IgG decline.

Reliance on the detection of N-IgG antibodies should not be used exclusively for determining long-term seroprevalence.

## Figures and Tables

**Figure 1 vaccines-10-01532-f001:**
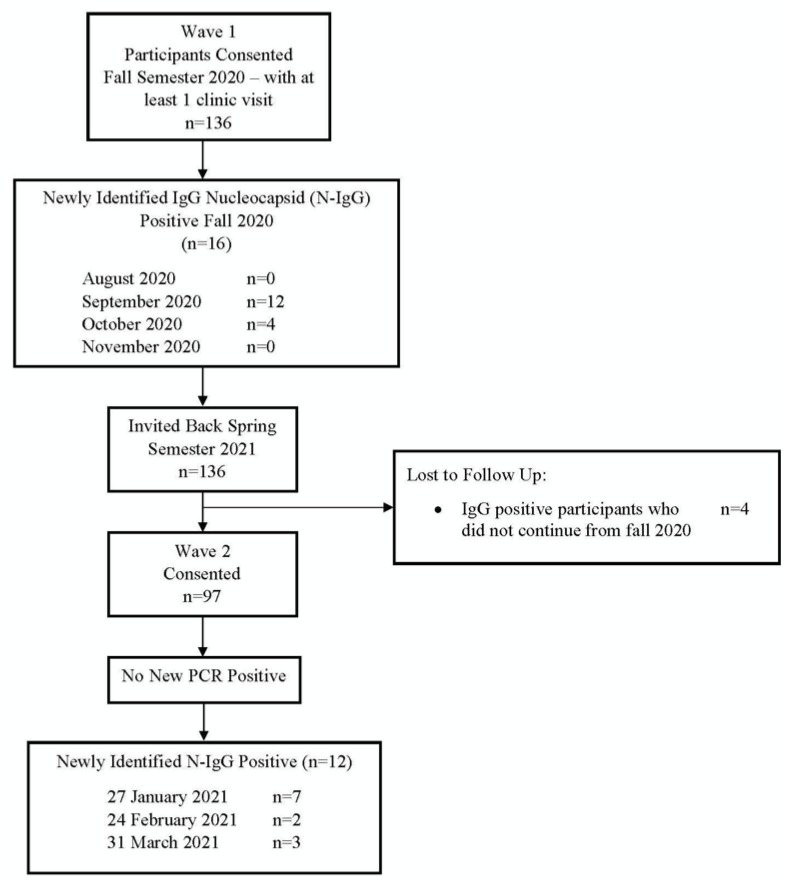
Participant flow between fall 2020 and spring 2021.

**Figure 2 vaccines-10-01532-f002:**
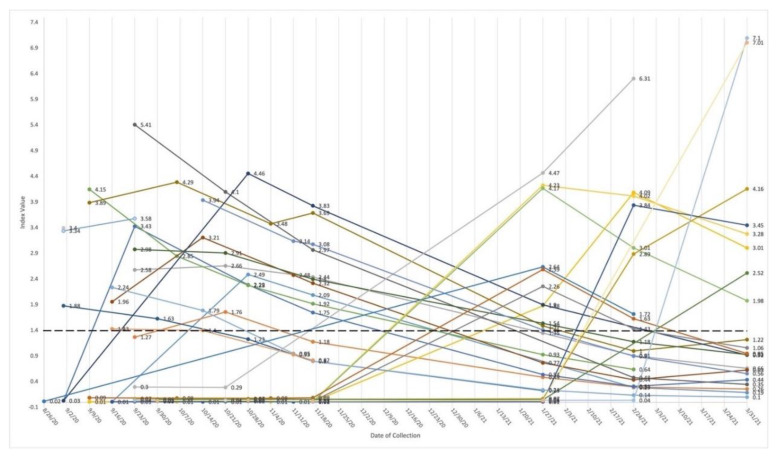
Change in quantity of antiN-IgG for 28 young adults with at least one N-IgG positive result, 26 August 2020–31 March 2021. Each individual is represented by a colored line from first testing date to last testing date. Number at dot is index value. An index value greater than or equal to (≥) 1.4 is positive (bold dotted line). The chemiluminescent microparticle immunoassay (CMIA) captures antibodies reactive with the Nucleocapsid (N) protein and specifically detects IgG only. This qualitative assay reports a ratio of luminescence between sample and calibrator (the S/C index) (Abbott Architect i2000 2). One N-IgG positive participant enrolled and withdrew in September (2 data points); one participant had one study visit in September 2020 (1 data point). Three participants tested positive at last study visit date (1 data point). The remaining 23 provided N-IgG measures across multiple weeks.

**Figure 3 vaccines-10-01532-f003:**
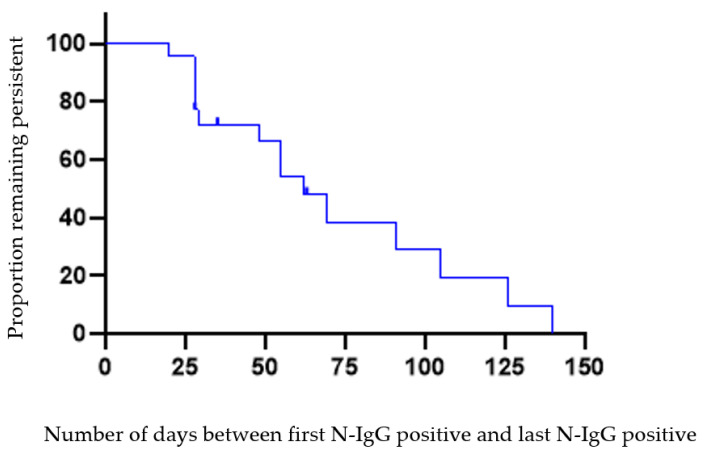
Loss of anti-N IgG persistence among 22 young adults. Decline in positive N-IgG status over 140 days (*n* = 22), between participant’s initial date of enrollment in fall semester 2020 until 31 March 2021. 68.1% had decline of N-IgG over 140 days. *X*-axis is days between 0 to 140 days of N-IgG persistence by end of follow-up. *Y*-axis is proportion of people remaining with IgG persistence. Seven (censored) had persistent detectable N-IgG; fifteen did not. Persistence is defined as number of days between first positive and last positive IgG nucleocapsid result. This graph does not reflect number of days between date of first positive PCR test positive and date of first IgG positive. One observation in Table 1, column 3 was excluded from Figure 3 due to withdrawal from participation in September 2020.

**Figure 4 vaccines-10-01532-f004:**
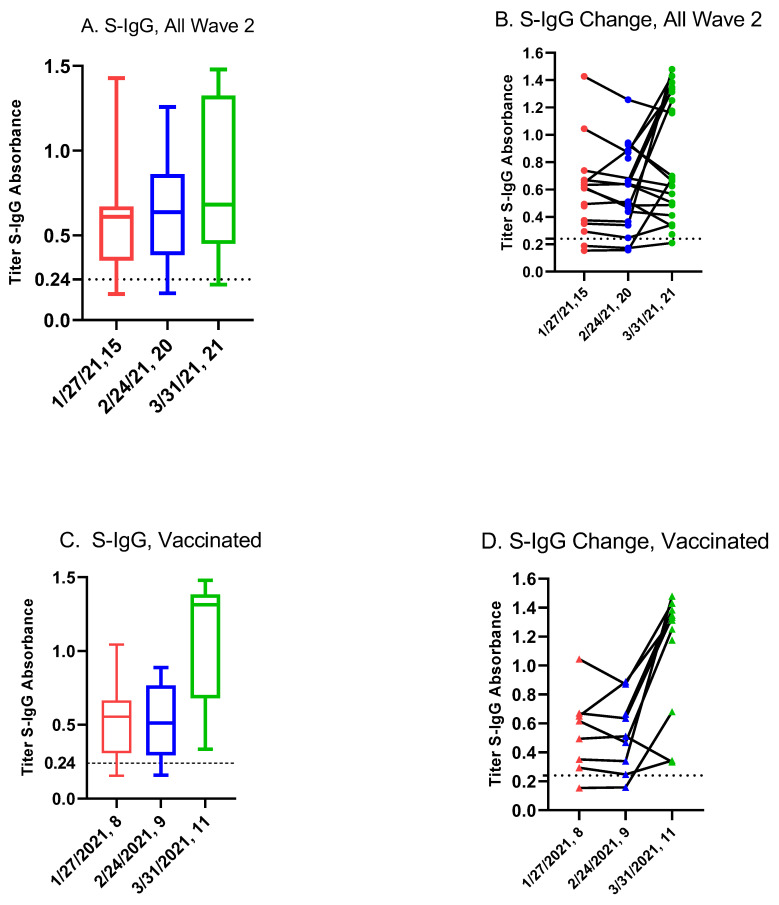
Anti-spike (S-IgG) change at three time points (January, red; February, blue; March green, respectively), and number of student sera at each time point. (**A**) Box and whisker plot by collection date for all students. (**B**) Line graph for change in S-IgG titer at three time points. (**C**) Box and whisker plot among vaccinated students that are a subset of graph (**A**). (**D**) Line graph for change in S-IgG among vaccinated students that are a subset of graph (**B**). One student was first vaccinated on 27 January 2021 and ten were first vaccinated after 2 March 2021.

**Table 1 vaccines-10-01532-t001:** Number of days of persistence and loss for N-IgG seropositivity in 23 young adults with greater than one positive N-IgG result, 1 September 2020 through 31 March 2021 ^1^.

Number of Days	PCR Positive Test to Initial N-IgG Positive (*n* = 19) ^2^	N-IgG Persistence *n* = 23 ^4^	Loss of Persistence ^5^ *n* = 15 ^6^
Mean, SD	21.21 (14.76)	54.30 (33.51)	82.33 (44.16)
Median	19	48	63
Range	0–54 3	20–140	28–154

SD, standard deviation. ^1^ Serology was offered approximately every 30 days from date of first collection, excluding December 2020. ^2^ Three participants were missing date of positive PCR test. One participant who self-reported PCR positive and tested IgG positive in September 2020 returned on 27 January 2021. This outlier was removed from Table 1 leaving 19 PCR-positive students; column 2 with participant’s values (days) were: mean, SD: 27.55 (31.78), median: 19.5, and range: 0–148 days. ^3^ Two tested PCR positive and IgG positive on same day; the number of days between PCR positive and IgG positive was zero. ^4^ IgG persistence is defined as number of days between date of initial positive N-IgG and date of last positive N-IgG. ^5^ Loss of IgG (days from initial IgG positive to first negative IgG result (S/C ratio < 1.4). ^6^ Excludes 8 with no decline of persistence.

## Data Availability

Data supporting findings are available from the corresponding author upon reasonable request. The data are not publicly available because publications are underway.

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
