# Peer review of "Waning of SARS-CoV-2 Seropositivity among Healthy Young Adults over Seven Months"

_vaccines, 2022, doi:10.3390/vaccines10091532_

Round 1
Reviewer 1 Report
The manuscript by Lea and colleagues examines the waning of SARS-CoV-2 seropositivity in young adults with time. The authors examine the IgG responses against the nucleocapsid antigen over a several months and observed that responses decreased within five months. Overall, the experiments were well thought out and the manuscript was well written. The manuscript adds to our knowledge of seroconversion and reversion with this important pathogen. My comments are mostly grammatical and are listed below.
Line 59-60: Limited longitudinal evidence exists on seroprevalence conversion and reversion among healthy young adults enrolled at university. The sentence would read better as “Limited longitudinal evidence exists on seroconversion and reversion among healthy young adults enrolled at university.
Line 60: At the end of this line, “enrolled at university” should be corrected to “enrolled at a university.”
Line 166-167: “participatns” should be corrected to participants
Figure 3, y axis: “Proportion remaining Persistent” should be “Proportion remaining persistent.”
Line 226: 48% should be changed to Forty-eight %.
Figure 4: The first word “anti-“ should be changed to “Anti-.”
Line 248: The word “thee” should be corrected to “three.”
Line 252-253: The statement “or the fact that it is expressed both on the virus particle and on the surface of infected cells” should be revised as studies have shown that the envelope (E) protein and the matrix (M) proteins can prevent the S from trafficking to the cell surface.
Line 278: 46% should be corrected to Forty-six %.
Line 279: The word “symotmatic” should be corrected to “symptomatic.”
Reviewer 2 Report
This paper is fairly well written with good details, references, and reported methods.
Some general writing problems throughout the manuscript.
There are some formal grammatical structures with some sentences that should be corrected to increase the clarity and strength of topic sentences. The following topic sentences (at the beginning of new paragraphs) should be changed and made into a direct statement (from an indirect statement) by moving the subject to the beginning of the sentence and putting the proposition and/or dependent clauses at the end of the sentence. This more direct statement makes for a much stronger and clearer topic sentence at the beginning of new paragraphs and better defines what the paragraph is about.
Instances of this problem are as follows (Lines):
L 38, L 43, L 97, L 173, L 190, L 199, L 223, L 272, L 306
Results
Figures 1-4 have figure descriptions imbedded (as part of the figure image). Please remove the figure description from the figure and place it in separate text below the figure.
Figures 2 and 3 are of low resolution (blurred and fuzzy); please increase the quality of these figures by raising the resolution to 300 dpi (or higher) so that numbers and lines are clear and readable.
Discussion
Good comments of your study results relative to previous published articles. Add to this (if possible) by including additional very recent articles published in 2022 to make the paper even more current (if such published pertinent articles exist).
Conclusions
The conclusions are quite brief. Please expand to include any implications of your results relative to potential applications of this information and list possible suggestions of how this new information (from your study) might but useful and appropriate for making improved changes in healthcare policies and messaging to the public.
Reviewer 3 Report
It is not stated what types of vaccines were used and whether there are differences in the response to the use of mRNA and attenuated vaccines?
Have there been any foreign students who have been vaccinated with any of the vaccines that are not accepted in America?
Has anyone been tested within 4 weeks of the first or second dose of the vaccine?
Round 2
Reviewer 2 Report
Much improved formatting of Figure presentations and inclusion of significance of findings for practical applications. The resolution and quality of Figure 2 could still be improved.
Reviewer 3 Report
/